# IG-NET: IMAGE-GOAL NETWORK FOR OFFLINE VISUAL NAVIGATION ON A LARGE-SCALE GAME MAP

## ABSTRACT

Navigating vast and visually intricate gaming environments poses unique challenges, especially when agents are deprived of absolute positions and orientations during testing. This paper addresses the challenge of training agents in such environments using a limited set of offline navigation data and a more substantial set of offline position data. We introduce the *Image-Goal Network* (IG-Net), an innovative solution tailored for these challenges. IG-Net is designed as an image-goal-conditioned navigation agent, which is trained end-to-end, directly outputting actions based on inputs without intermediary mapping steps. Furthermore, IG-Net harnesses position prediction, path prediction and distance prediction to bolster representation learning to encode spatial map information implicitly, an aspect overlooked in prior works. Our experiments and results demonstrate IG-Net's potential in navigating large-scale gaming environments, providing both advancements in the field and tools for the broader research community.

## 1 INTRODUCTION

Visual navigation, the act of autonomously traversing and understanding environments based on visual cues, has been at the forefront of robotics and artificial intelligence research (Shah et al., 2021; 2023; Kwon et al., 2021). The ability to navigate is a fundamental skill for agents, making it applicable in a wide range of scenarios, from virtual gaming environments to real-world robotic applications. The challenge, however, lies in the complexity and variability of these environments, especially when the scale is vast and the available data is limited. In this work, we consider the ShooterGame environment constructed by Unreal Engine with realistic visual dynamics, as illustrated in Figure 1, which spans 10421.87 $m^2$ across multiple floors, representing a scale approximately 50-100 times larger than preceding navigational environments.

Such a large-scale environment presents intricate navigational challenges, especially when the agent's access to navigation data is restricted to offline modes, in which the navigation data needs (human) experts to control the agent to reach the goal. On the other hand, we can use some unlabeled data (without action and noncontinuous, which can be accessed by random sample) to enhance the model training, obeying the rule of weakly supervised learning (Zhou, 2017; Gong et al., 2022). In such a case, during the training phase, the agent can access only a limited number of navigation (supervised) data, comprising positions and images, supplemented by a (unsupervised) dataset containing merely positions and images. In contrast, the testing phase imposes further restrictions, allowing the agent access solely to the current observational image and the goal image. The primary objective of this research is to navigate the agent proficiently to reach the predefined goal, relying exclusively on image observations. The observational data is limited to a 90-degree camera, posing considerable challenges compared to the conventional 360-degree camera observations, making it imperative to devise robust solutions to navigate efficiently with purely image-based observations during the testing phase and training exclusively with offline data (Al-Halah et al., 2022).

To mitigate the challenges intrinsic to such constrained and expansive environments, we propose the *Image-Goal Network* (IG-Net), an end-to-end solution specifically designed for large-scale visual navigation tasks. This network amalgamates visual and positional information to guide the agent towards its goal effectively. Besides, explicitly building a map for navigating on such a large-scale environment is quite challenging, while we still need to fuse the map knowledge into the navigation model training. To this end, we incorporate spatial information, representing the positional infor-

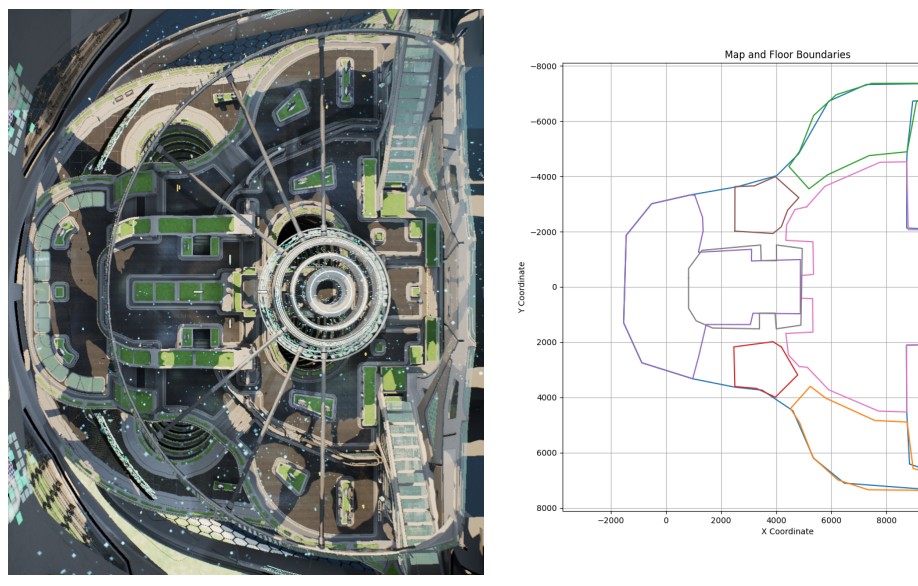

(a) A bird view of ShooterGame  (b) A sketch of the ShooterGame

Figure 1: A bird view and a sketch of usable space of ShooterGame.

mation of each image implicitly for a better representation, in enhancing the agent's navigational capabilities. Our experiments, conducted within the large-scale ShooterGame map, substantiate the effectiveness of the proposed IG-Net in navigating through extensive and intricate environments using offline, image-based data. The results demonstrate significant advancements in visual navigation, opening avenues for further research and development in autonomous navigation in large-scale, complex environments.

## 2 RELATED WORKS

Image-goal visual navigation within large-scale maps, particularly when devoid of absolute positions during testing and online interactions during training, poses a profound challenge addressed by numerous research endeavors. ViNG (Shah et al., 2021) is an exemplar model in this realm, predicting steps and accessibility to a target while generating waypoints, utilizing a controller, constructing trees, and planning paths via a weighted Dijkstra algorithm. The model's distance function is adeptly trained through temporal difference learning or supervised learning. Contrastingly, BADGR (Kahn et al., 2021), and subsequent developments (Hahn et al., 2021), employ self-supervised learning paradigms focusing on end-to-end learning from real-world data, devoid of simulations or human demonstrations. The emphasis on "last-mile navigation" is brought forth by another study (Wasserman et al., 2022), accentuating goal discovery and exploitation post-object identification.

Advancements in topologist-based methodologies have also been noteworthy. The Topological Semantic Graph Memory (Kim et al., 2022) model, utilizing depth cameras, constructs graphs based on images or objects and avoids reliance on positional information, employing a cross-graph mixer for updates. Similarly, Visual Graph Memory (Kwon et al., 2021) leverages landmark-based topological representations for zero-shot navigation in novel environments, and the Neural Topological SLAM (Chaplot et al., 2020) updates graphs through nodes representing 360-degree panoramic views based on agent observations. The visual representation R3M (Nair et al., 2022) demonstrates the potential of data-efficient learning for downstream robotic manipulation tasks using pre-trained visual representations on diverse human video data. Moreover, PACT (Bonatti et al., 2022) introduces a generative transformer-based architecture that builds robot-specific representations from robot data in a self-supervised fashion, evidencing enhanced performance in tasks such as safe navigation, localization, and mapping compared to training models from scratch.

On the other hand, some pretrained networks were proposed to solve visual navigation across a variety of environments. PIRLNav (Ramrakhya et al., 2023) addresses the challenges of designing

embodied agents that can interact effectively with their environment and humans by combining behavior cloning (BC) and reinforcement learning (RL) to overcome the limitations inherent to each method individually. Also, Majumdar et al. (2023) conducts an extensive empirical study focusing on the design of an artificial visual cortex, aimed at enabling an artificial agent to convert camera input into actions. Our proposed method shares the idea of representation learning of pretrained network methods, but the problem settings are different.

Our approach, the *Image-Goal Network* (IG-Net), stands distinctively apart from these methods. Unlike the aforementioned works, IG-Net is meticulously designed to navigate significantly larger maps, where interactions are strictly confined to offline data, and there is an absence of online interactions with the environment or position signals during testing. Furthermore, our emphasis on amalgamating visual and positional information to guide the agent effectively through extensive environments, as evidenced in the ShooterGame environment, presents a novel perspective in addressing visual navigation challenges.

## 3    PROBLEM SETTING

In this study, we tackle the intricate problem of visual navigation within a large-scale map, specifically, within the ShooterGame environment. This game environment is significantly expansive, featuring a 10421.87 m$^2$ map with multiple levels, making it approximately 50-100 times larger than previously utilized navigation environments, as illustrated in Table 1.

| Environment | Gibson | SUNCG | Matterport3D | ShooterGame |
|---|---|---|---|---|
| Coverage of One Map ($m^2$) | 368.88 | 127.13 | 517.78 | 10421.87 |
| Dynamic Objects | ✗ | ✗ | ✗ | ✓ |
| Pure RGB without depth | ✗ | ✗ | ✗ | ✓ |
| No Panoramic 360 camera view | ✗ | ✗ | ✗ | ✓ |

Table 1: Comparison of ShooterGame with previous navigation environments, including Gibson (Xia et al., 2018), SUNCG (Song et al., 2017) (hand designed synthetic), Matterport3D (Chang et al., 2017), and MINOS (Savva et al., 2017). The coverage of one task of ShooterGame is 50-100 times bigger than previous ones.

**Environment.** ShooterGame is a quintessential representation of a PC multiplayer first-person shooter by Unreal Engine 4, providing a robust framework that includes diverse weapon implementations, game modes, and a simplistic front-end menu system, with observations further constrained to 90-degree camera views [1]. This restriction augments the challenge compared to preceding 360-degree camera observations as in Figure 2. From the figure we can also observe that the distant craft, clouds, sunlight, and even walls change dynamically over time, resulting in different observations of the same position and angle at different moments in time, which renders the navigation within this environment a complex endeavor. The lack of depth information also poses unique implementations and challenges for navigation tasks.

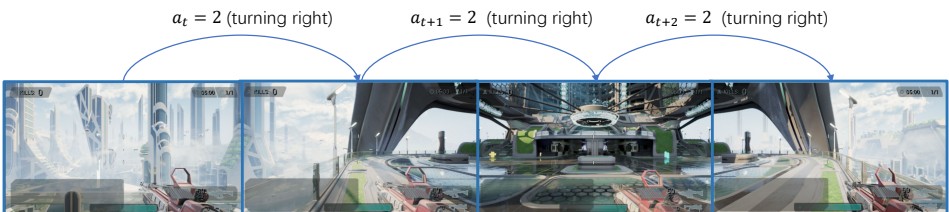

Figure 2: Four image observations from one fixed position with different view angles. Each observation only contains a 90-degree camera view, which can be more challenging than previous 360-degree-view environments.

**Offline Dataset.** We collect a set of navigation data to train our models. At each step, the agent is given the current 90-degree camera observation $o_t$ and the target 90-degree observation $o_{tar}$. A human-expert action $a_t$ is executed which leads to next observation. The trajectory ends when

---

[1]A public video demo: https://www.youtube.com/watch?v=xdS6asajHAQ

agent navigates to target. Global positions and rotations $\boldsymbol{p}_t = (x_t, y_t, z_t, \theta_t)$ are also collected but only used for auxiliary tasks and visualizations, and the magnitude of coordinates $(x_t, y_t, z_t)$ is normalized to zero-mean and unit-variance. Each trajectory in our dataset is represented as:

$$\tau^i = \{\boldsymbol{o}^i_{tar}, \boldsymbol{o}^i_0, \boldsymbol{p}^i_0, \boldsymbol{a}^i_0, \ldots, \boldsymbol{o}^i_{T^i}, \boldsymbol{p}^i_{T^i}, \boldsymbol{a}^i_{T^i}\}, \tag{1}$$

in which $T^i$ denotes the length of trajectory $i$; $\boldsymbol{o}^i_t$ denotes the image observation in $i$-th trajectory at time step $t$ with image size $224 \times 224 \times 3$; $\boldsymbol{o}_{tar}$ denotes the goal observation; $\boldsymbol{p}^i_t$ denotes the current position and current rotation in $i$-th trajectory at time step $t$ with a tuple $(x^i_t, y^i_t, z^i_t, \theta^i_t)$ coordinate in the coordinate system; $a^i_t \in 0, 1, 2$ denotes the current action for moving forward, turning left, and turning right, respectively. A total of $N = 200$ trajectories are collected for training following this manner. Each trajectory takes less than 2 minutes to collect and lasts an average of 75 steps.

We collect additional position data that includes the positions-images of the map. We uniformly sample a set of $M = 2000$ points in the map, where each point is represented as the observation-position-rotation pair $\{(\boldsymbol{o}^i, \boldsymbol{p}^i) | 1 \leq i \leq M\}$, where $\boldsymbol{p}^i = (x^i, y^i, z^i, \theta_i)$. This position dataset is used solely for representation learning in the visual navigation training.

**Training and Testing Phases.** The agent is constrained to offline data during training, incorporating a limited number of navigation data and position-image data without any interactions with the environment. The testing phase restricts the agent's access to the current observation image $\boldsymbol{o}$ and the goal image $\boldsymbol{o}_{tar}$ only. The target of the task is to navigate the agent from an initial position to a goal position.

The problem of visual navigation within the large-scale and detailed ShooterGame environment of Unreal Engine poses significant challenges due to the limited availability of offline data and restrictive observations. In the following sections we will introduce our proposed solution, Image-Goal Network (IG-Net), which navigates through this environment, emphasizing the importance of spatial information in visual navigation tasks. The promising results from preliminary experiments indicate the potential of our approach in addressing the challenges inherent in sophisticated game environments like ShooterGame.

## 4 PROPOSED METHOD: IG-NET

Addressing the intricacies and challenges imposed by large-scale, constrained environments necessitates a methodological paradigm shift. The Image-Goal Network (IG-Net) is developed to cater to the nuanced demands of visual navigation within expansive and intricate gaming terrains.

### 4.1 FOUNDATION PRINCIPLES

Given the extensive scale of the map and the unavailability of online interaction with the environment, constructing a model explicitly based on the map, such as topological methods (Kim et al., 2022), is not feasible. Accordingly, IG-Net is proposed with distinct properties to navigate proficiently within such constrained settings:

- **Image-Goal-Conditioned Behavior:** IG-Net fundamentally operates as an image-goal-conditioned navigation agent. In inference, it consumes an image and utilizes it as a navigational goal, shaping its navigation behavior correspondingly.
- **End-to-End Training:** Distinct from conventional methodologies, which prioritize constructing a comprehensive map or graph of the environment initially, IG-Net adopts end-to-end training. This approach allows IG-Net to directly interpret inputs and output actions, bypassing intermediary mapping processes.
- **Enhanced Representation Learning through Position and Navigation Information Prediction:** IG-Net utilizes the nuanced capabilities of position and navigation information prediction to refine representation learning, a domain relatively untouched in preceding studies. It employs spatial information prediction to enhance the agent's internal environmental depiction.
- **Incorporation of Auxiliary Tasks:** A variety of auxiliary tasks are integrated, including local and global path planning and navigation distance prediction, to fortify visual navigation capabilities.

### 4.2 LOSS FUNCTIONS

To optimize IG-Net, we devise a conglomerate of loss functions, each catering to different aspects of navigation, such as positional accuracy, trajectory optimization, and alignment with the goal state. The losses ensure the coherent learning of representations and navigational strategies, which are crucial for effective navigation in complex environments. We denote the parameter of IG-Net as $\theta$, and each function of IG-Net parameterized by $\theta$ is detailed in the following sections.

**Relative Position Prediction.** The relative position prediction design allows IG-Net to learn to inherit spatial representation given camera views on the map. Given any two states represented by $(\boldsymbol{o}_1, \boldsymbol{p}_1)$ and $(\boldsymbol{o}_2, \boldsymbol{p}_2)$, we compute the relative position and orientation of these two states as:

$$\mathrm{relative}(\boldsymbol{p}_2, \boldsymbol{p}_1) = \left((x_2 - x_1, \quad y_2 - y_1, \quad z_2 - z_1)\, R(-\theta_1)^T, \theta_2 - \theta_1\right), \qquad (2)$$

where $R(\theta)$ is the rotation matrix for angle $\theta$. Qualitatively, $\mathrm{relative}(\boldsymbol{p}_2, \boldsymbol{p}_1)$ reflects the position and rotation of $\boldsymbol{o}_2$ in the egocentric coordinates of $\boldsymbol{o}_1$. Given a pair of images, IG-Net is able to predict the relative position of the images, and the following loss function is used for relative position prediction in IG-Net:

$$L^{\mathrm{relative}}(\boldsymbol{o}_1, \boldsymbol{p}_1, \boldsymbol{o}_2, \boldsymbol{p}_2) = L^{\mathrm{pos\_angle}}(f_\theta^{\mathrm{relative}}(\boldsymbol{o}_1, \boldsymbol{o}_2), \mathrm{relative}(\boldsymbol{p}_2, \boldsymbol{p}_1)), \qquad (3)$$

where

$$L^{\mathrm{pos\_angle}}((x_1, y_1, z_1, \theta_1), (x_2, y_2, z_2, \theta_2))$$
$$= \|(x_2 - x_1, y_2 - y_1, z_2 - z_1, \cos(\theta_2) - \cos(\theta_1), \sin(\theta_2) - \sin(\theta_1))\|_2^2$$

evaluate how the predicted relative positions and rotations are close to the ground truth relative positions and rotations.

One advantage of relative position is that any data with position information can be leveraged. We use a mixture of position offline data and navigation offline data for training the relative position prediction, detailed later in this section.

**Absolute Position Prediction.** We additionally use IG-Net to predict the absolute position and rotations given a camera view, serving as an additional auxiliary loss for IG-Net. Given one state represented by $(\boldsymbol{o}_1, \boldsymbol{p}_1)$, the following loss function is used for training IG-Net is given by:

$$L^{\mathrm{absolute\_pos}}(\boldsymbol{o}_1, \boldsymbol{p}_1) = L^{\mathrm{pos\_angle}}(f_\theta^{\mathrm{absolute\_pos}}(\boldsymbol{o}_1), \boldsymbol{p}_1). \qquad (4)$$

We also use a mixture of offline navigation data and position data for training IG-Net.

**Navigation distance prediction.** For the navigation distance prediction task, IG-Net is given a pair of states represented by image observations, and learns to predict the total distance that takes the agent to navigate from the first state to the second state. When the loss is optimized, the network captures the connectivity between different states in the map. Given a trajectory $\tau = (\boldsymbol{o}_{tar}, \boldsymbol{p}_{tar}, \boldsymbol{o}_0, \boldsymbol{p}_0, \boldsymbol{a}_0, \ldots, \boldsymbol{o}_T, \boldsymbol{p}_T, \boldsymbol{a}_T)$ in the offline navigation dataset, we let $\boldsymbol{o}_{T+1} = \boldsymbol{o}_{tar}$ and $\boldsymbol{p}_{T+1} = \boldsymbol{p}_{tar}$ define the navigation distance between $\boldsymbol{o}_i, \boldsymbol{o}_j, i \leq j$ as follows:

$$\mathrm{nav\_distance}(\boldsymbol{o}_i, \boldsymbol{o}_j, \tau) = \sum_{k=i}^{j-1} \|(x_k - x_{k+1}, y_k - y_{k+1}, z_k - z_{k+1})\|_2^2. \qquad (5)$$

Given a pair of states in the offline navigation dataset, IG-Net predicts the navigation distance between them. The loss function for training IG-Net is given by:

$$L^{\mathrm{nav\_distance}}(\boldsymbol{o}_i, \boldsymbol{o}_j, \tau) = \left[f_\theta^{\mathrm{nav\_distance}}(\boldsymbol{o}_i, \boldsymbol{o}_j) - \mathrm{nav\_distance}(\boldsymbol{o}_i, \boldsymbol{o}_j, \tau)\right]^2 \qquad (6)$$

**Navigation path prediction.** Given a pair of states represented by image observations, IG-Net learns to construct the spatial navigation path between them, serving as a path-planning auxiliary loss for IG-Net. For the local path prediction in IG-Net, the predicted path is the $N_{path} = 5$ next consecutive steps in the navigation trajectory; for the global path prediction in IG-net, the predicted

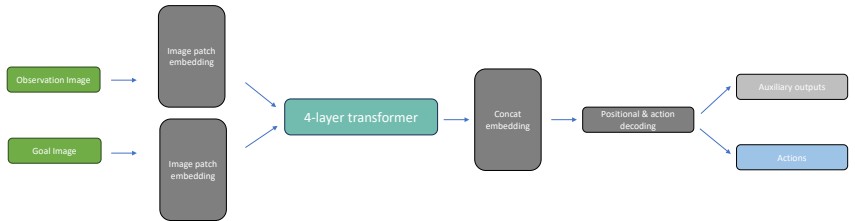

Figure 3: IG-Net architecture illustration, in which auxiliary outputs denote predictions on map positions & orientations; ego positions & orientations; local & global paths.

path is the $N_{path} = 5$ intermediate relative positions, where the intermediate points are equally spaced in time from current time $t$ to the total path length $T$.

Formally, we define the local and global timesteps as

$$S^{\text{local}}(t, \tau) = (\min(t+1, T), \min(t+2, T), \ldots, \min(t+N_{path}, T)), \tag{7}$$

$$S^{\text{global}}(t, \tau) = \left(t + \lfloor \frac{T-t}{N_{path}+1}\rfloor, t + \lfloor \frac{2(T-t)}{N_{path}+1}\rfloor, \cdots, t + \lfloor \frac{N_{path}(T-t)}{N_{path}+1}\rfloor\right). \tag{8}$$

We define the local and global path at timestep $t$ in the trajectory $\tau$ as

$$\text{local\_path}(\boldsymbol{o}_t, \tau) = \left(\text{relative}(S^{\text{local}}(t, \tau)_1, \boldsymbol{p}_t), \cdots, \text{relative}(S^{\text{local}}(t, \tau)_{N_{path}}, \boldsymbol{p}_t)\right), \tag{9}$$

$$\text{global\_path}(\boldsymbol{o}_t, \tau) = \left(\text{relative}(S^{\text{global}}(t, \tau)_1, \boldsymbol{p}_t), \cdots, \text{relative}(S^{\text{global}}(t, \tau)_{N_{path}}, \boldsymbol{p}_t)\right). \tag{10}$$

Finally, the training loss on local and global paths for IG-Net is defined as:

$$L^{\text{local\_path}}(\boldsymbol{o}_t, \tau) = \sum_{k=1}^{N_{path}} \left[L^{\text{pos\_angle}}(f_\theta^{\text{local\_path}}(\boldsymbol{o}_t, \boldsymbol{o}_{tar})_k, \text{local\_path}(\boldsymbol{o}_t, \tau)_k)\right], \tag{11}$$

$$L^{\text{global\_path}}(\boldsymbol{o}_t, \tau) = \sum_{k=1}^{N_{path}} \left[L^{\text{pos\_angle}}(f_\theta^{\text{global\_path}}(\boldsymbol{o}_t, \boldsymbol{o}_{tar})_k, \text{global\_path}(\boldsymbol{o}_t, \tau)_k)\right]. \tag{12}$$

**Action Loss.** Besides all the above auxiliary loss, we use an additional action loss for training IG-Net to generate navigation actions. Given one current image and one goal image, we train IG-Net to predict the current action. The action prediction head of IG-Net is trained with behavior cloning loss:

$$L^{\text{action}}(\boldsymbol{o_t}, \tau) = \text{cross\_entropy}(f^{\text{action}}(\boldsymbol{o}_t, \boldsymbol{o}_{tar}), a_t) \tag{13}$$

**Training Loss.** We add all the auxiliary loss and action prediction loss as a single loss function to train IG-Net. We use $w = 1.0$ for each loss term in our experiment.

**Sampling in position and navigation dataset for training IG-Net.** All the position prediction losses are trained with both position and navigation datasets. In contrast, navigation distance and path prediction loss rely solely on the navigation dataset. In our experiment, we sample the position dataset with $p_{pos} = 0.4$ probability and the navigation dataset with $1 - p_{pos} = 0.6$ probability. When sampled on the position dataset, the navigation distance and path prediction loss are masked in training. Our approach enables leveraging both the position and navigation datasets for training different auxiliary tasks without losing data efficiency.

## 4.3 ARCHITECTURAL DESIGN

IG-Net integrates a transformer-based structure, specifically tailored for navigation tasks, with a forward process described as follows:

1. **Image Encoding:** A pretrained Masked Auto-Encoder (MAE) is employed for encoding the current and goal images independently, ensuring a rich representation of visual information.

2. **Embedding Concatenation:** The encoding embeddings procured from the first step are concatenated to form a unified representation, encompassing both current state and goal state information.
3. **Positional and Action Decoding:** Utilizing position, path, distance, and action decoders, the network predicts corresponding positional information and navigational actions, leveraging the concatenated embeddings.

An illustration of the architecture of IG-Net is shown in Figure 3.

### 4.4 TRAINING AND INFERENCE

During training, IG-Net is exposed to a plethora of offline navigation data, enriched with positional and visual information. The network learns to intertwine visual and spatial cues to formulate robust navigational policies. In inference, the network, confined to current observational and goal images, generates actions to navigate the agent proficiently toward the predefined goal, overcoming the constraints imposed by limited observational data and expansive environments. The training hyperparameters for IG-Net are detailed in Appendix A.1.

## 5 EXPERIMENT

### 5.1 EXPERIMENT SETTING

We evaluate our models in three levels of difficulties according to the euclidean distance to goal ($D$) at the start of the episode. We run 50 episodes under each setting with a maximum of 200 steps per episode. Success is marked by agent locating within a fixed range of the goal, regardless of its orientation.

Our model is compared against Visual-Graph-Memory (VGM), a powerful end-to-end visual navigation algorithm. VGM aims to progressively build and update a topological graph representation of the environment during execution purely based on visual information. We deliberately select VGM from various baselines (Shah et al., 2021) (Kahn et al., 2021) (Kim et al., 2022) to compare an end-to-end *explicit* graph construction method with our *implicit* map features learning through auxiliary tasks. We train VGM variants with 90 FoV and mask out the depth input. When evaluating the "loaded" variants, we pre-load the model with nodes from a *training* episode. Other details can be found in A.2.

### 5.2 EVALUATION METRICS

Three metrics are used in this paper: success rate (SR), success weighted by path length (SPL), and distance decrement rate (DDR).

SPL measures the efficiency of navigation and can be written as $SPL = \frac{1}{N} \sum_{i=1}^{N} S_i \frac{d_i}{max(d_i, p_i)}$ where $S_i = 1$ if navigation is successful and $S_i = 0$ otherwise. $N$ is the total number of evaluation episodes, $d_i$ is shortest distance to target, and $p_i$ is the actual trajectory length. Since optimal geodesic distance is not available in ShooterGame, we use Euclidean distance for $d_i$.

DDR measures the closest distance achieved between the agent and the target towards to and can be written as $DDR = \frac{d_0 - d_{min}}{d_0}$ where $d_0$ represents the initial Euclidean distance from agent to goal and $d_{min}$ is the minimum Euclidean distance throughout the navigation trajectory.

### 5.3 RESULTS

The overall results are presented in Table 2. Under the easiest setting, IG-Net's success rate outperforms the best VGM variant by a margin of $69\%$ (from 0.32 to 0.54) and is $71\%$ more efficient in SPL (from 0.14 to 0.24). More remarkably, IG-Net achieves a reasonable success rate of 0.24 to 0.26 under more challenging settings whereas variants of VGM almost completely fail the tasks with success rates consistently below 0.1, despite our tuning efforts.

Moreover, results show that training IG-Net with auxiliary tasks significantly improves performance in both success rate and navigation efficiency. Therefore, we conclude the learning objectives proposed in 4.2 help IG-Net to establish an implicit and transferable understanding of the map.

| Difficulty | $1500 < D < 4000$ | | | $4000 < D < 8000$ | | | $D > 8000$ | | |
|---|---|---|---|---|---|---|---|---|---|
| Metric | SR | SPL | DDR | SR | SPL | DDR | SR | SPL | DDR |
| VGM | 0.24 | 0.09 | 0.40 | 0.00 | 0.00 | 0.30 | 0.00 | 0.00 | 0.27 |
| VGM-Tuned | 0.20 | 0.09 | 0.39 | 0.08 | 0.05 | 0.39 | 0.00 | 0.00 | 0.32 |
| VGM-Loaded | 0.32 | 0.14 | 0.49 | 0.04 | 0.02 | 0.26 | 0.00 | 0.00 | 0.27 |
| VGM-Tuned-Loaded | 0.20 | 0.08 | 0.32 | 0.04 | 0.02 | 0.36 | 0.08 | 0.05 | 0.42 |
| IG-Net | **0.54** | **0.24** | **0.75** | **0.26** | **0.17** | **0.65** | **0.24** | **0.15** | **0.75** |
| IG-Net (no auxiliary) | 0.18 | 0.09 | 0.36 | 0.14 | 0.08 | 0.42 | 0.00 | 0.00 | 0.44 |

Table 2: IG-Net experiment results. SR: success rate. SPL: success-weighted by path length. DDR: distance decrement rate.

## 5.4 CASE STUDY

### 5.4.1 VISUALIZATION OF NAVIGATION PATH OF IG-NET

To demonstrate IG-Net's proficiency in visual navigation, especially with purely visual inputs in complex environments such as ShooterGame, we present case studies depicting the navigation paths executed by IG-Net during the evaluation phase, as illustrated in Figure 4. From its initial position, IG-Net successfully executes its planning paths and executes low-level actions seamlessly, navigating through stairs and corridors while avoiding collisions with obstacles. These observations are consistent across various evaluation episodes, showcasing IG-Net's capability to navigate accurately towards the goal image and execute precise low-level navigational maneuvers to follow the correct path.

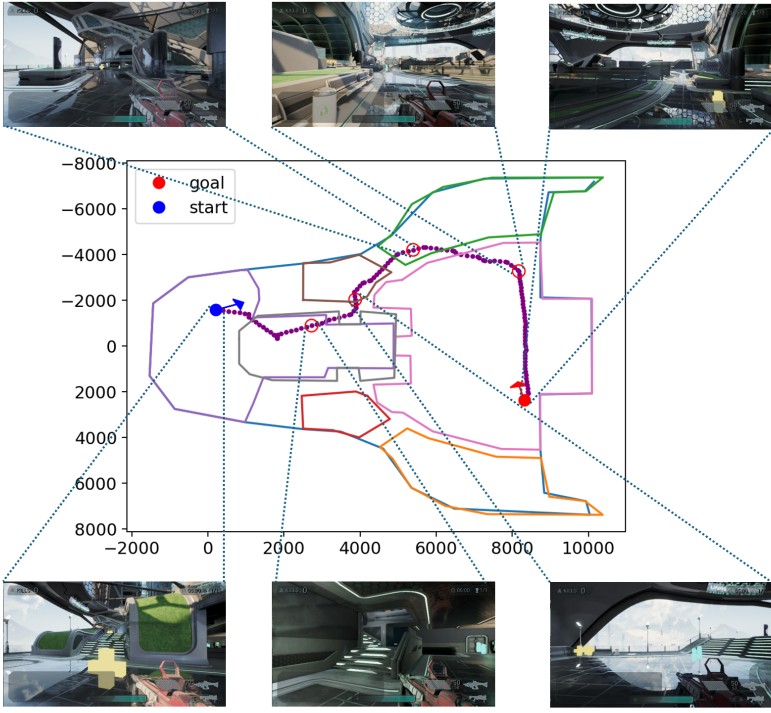

Figure 4: Illustration of IG-Net's navigation path in ShooterGame during evaluation. The bottom-left figure represents the agent's starting position, and the top-right figure displays the goal image, serving as input to IG-Net. Purple dots trace the path navigated by IG-Net, and red dots represent key frames in the navigation, with corresponding images visualized.

### 5.4.2 ROBUSTNESS OF IG-NET

To assess IG-Net's robustness, we conduct a case study introducing Gaussian noises, denoted as $n$, to the positions. We normalize of the position to zero-mean and unit-variance, and add a noise on all position training signals with standard derivation of $n$. Table 3 reveals that IG-Net maintains substantial performance even amidst high noise levels. Intriguingly, noise appears to enhance IG-Net's performance in challenging tasks ($D > 8000$), a phenomenon akin to utilizing noise to augment agents' exploration capability in RL scenarios (Eberhard et al., 2023; Plappert et al., 2018; Fortunato et al., 2018). This unexpected benefit opens up promising avenues for future enhancements to IG-Net's performance.

| Difficulty | $1500 < D < 4000$ | | | $4000 < D < 8000$ | | | $D > 8000$ | | |
|---|---|---|---|---|---|---|---|---|---|
| Metric | SR | SPL | DDR | SR | SPL | DDR | SR | SPL | DDR |
| IG-Net | **0.54** | **0.24** | **0.75** | **0.26** | **0.17** | **0.65** | 0.24 | 0.15 | **0.75** |
| IG-Net ($n = 0.1$) | 0.26 | 0.13 | 0.47 | 0.22 | 0.14 | 0.61 | 0.16 | 0.11 | 0.64 |
| IG-Net ($n = 0.2$) | 0.42 | 0.18 | 0.58 | 0.16 | 0.09 | 0.58 | **0.30** | **0.20** | 0.74 |
| IG-Net ($n = 0.4$) | 0.26 | 0.12 | 0.52 | 0.18 | 0.09 | 0.61 | 0.20 | 0.12 | 0.70 |

Table 3: Performance of IG-Net under different noise levels.

### 5.4.3 WHY VGM FAILS

VGM, along with several other methodologies (Kim et al., 2022), strives to represent environments using nodes and vertices, relying solely on visual information. Our findings suggest that in expansive gaming environments like ShooterGame, graph construction is prone to failure and necessitates meticulous hyperparameter tuning (refer to A.2). Moreover, the nodes in VGM often encompass only a minor section of the large-scale map, hindering the algorithm from utilizing prior map information to facilitate new navigation tasks.

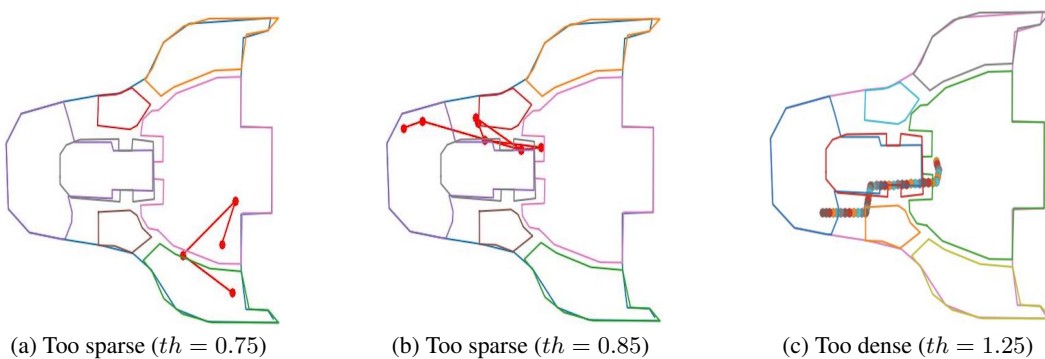

(a) Too sparse ($th = 0.75$)   (b) Too sparse ($th = 0.85$)   (c) Too dense ($th = 1.25$)

Figure 5: Illustration of failed VGM node construction under varying parameters.

Furthermore, VGM nodes often only cover a minor section of the large-scale map, which prevents the algorithm from leveraging prior map information to guide new navigation tasks.

### 5.5 ABLATION STUDY

In this section, we explore the auxiliary tasks' contribution to representation learning and the subsequent enhancement of IG-Net's navigation capabilities. The results are detailed in Table 4. It is evident that the absence of various auxiliary tasks leads to performance degradation to varying degrees. The IG-Net (no aux) variant, lacking all auxiliary losses, exhibits the most considerable performance decline. These results conclusively show that the designed auxiliary tasks significantly enrich IG-Net's representation and, consequently, elevate its navigation performance.

| Difficulty | $1500 < D < 4000$ | | | $4000 < D < 8000$ | | | $D > 8000$ | | |
|---|---|---|---|---|---|---|---|---|---|
| Metric | SR | SPL | DDR | SR | SPL | DDR | SR | SPL | DDR |
| IG-Net | **0.54** | **0.24** | **0.75** | 0.26 | **0.17** | **0.65** | 0.24 | 0.15 | **0.75** |
| IG-Net (no position) | 0.30 | 0.14 | 0.43 | **0.28** | 0.14 | 0.59 | 0.12 | 0.07 | 0.55 |
| IG-Net (no path and dist) | 0.38 | 0.17 | 0.58 | 0.26 | 0.14 | 0.62 | **0.30** | **0.20** | 0.66 |
| IG-Net (no auxiliary) | 0.18 | 0.09 | 0.36 | 0.14 | 0.08 | 0.42 | 0.00 | 0.00 | 0.44 |

Table 4: Ablation study on the impact of auxiliary losses.

## 6 CONCLUSION

In this study, we tackled the intricate challenges of visual navigation in expansive gaming environments with the introduction of the Image-Goal Network (IG-Net). IG-Net is a testament to the synergy of cutting-edge deep learning and specialized navigation strategies, emphasizing image-goal-conditioned behavior and the implicit encoding of spatial map information, a facet underexplored in preceding works. The network's proven adaptability and robustness in the expansive ShooterGame environment underscore its potential in navigating large-scale, visually rich domains using solely offline, image-centric data. The significant advancements delivered by IG-Net are not confined to enhancing visual navigation but extend to enriching representation learning, providing invaluable insights and tools for ongoing and future investigations in both virtual and real-world autonomous navigation applications. The foundational principles of IG-Net are poised to influence the development of more sophisticated navigation agents.

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

## REPRODUCIBILITY STATEMENT

We have attached IG-Net code with our submission. The authors are also committed to release the training and evaluation datasets upon paper being accepted.

## A    TRAINING DETAILS

### A.1    IG-NET PARAMETERS

The hyperpameters of training IG-Net is provided as follows:

| Param Name | IG-Net |
|---|---|
| Learning Rate | $5e^{-5}$ |
| Batch Size | 16 |
| N (Number of Navigation Trajectories) | 200 |
| M (Number of Position Data) | 2000 |
| Visual Dim | $224 \times 224 \times 3$ |
| Visual Backbone | MAE |
| Max Epoch | 100 |
| Action Dim | 3 |
| Path prediction length | 5 |
| Loss weights for each auxiliary loss | 1.0 |
| Loss weights action loss | 1.0 |

### A.2    VGM PARAMETERS

We mostly follow the default VGM training procedure by keeping the architecture of the model the same as in the original publication. To train under ShooterGame, we cut panoramic observations to 90 FoV and mask the depth input. To overcome the issue of having sparse nodes during training, we tune the $th$ parameter to different values to loosen node generation criteria. Finally, we train for a maximum of 250 epochs and choose the model checkpoint when validation loss achieves the lowest.

| Param Name | Default | Tuned |
|---|---|---|
| Learning Rate | $1e^{-4}$ | - |
| Batch Size | 4 | - |
| Th | 0.75 | 0.85 |
| Visual Dim | $64 \times 64 \times 3$ | - |
| Visual Backbone | ResNet-18 | - |
| Max Epoch | 250 | - |
| Action Dim | 3 | - |

## B    DATASET DETAILS

### B.1    TRAINING DATASET

All training navigation trajectories are collected by human experts. There are a total of 200 trajectories and each takes less than 2 minutes to collect. Human experts have prior experience with the game environment and are given additional information such as goal location on the map and distance to the goal to facilitate efficient collection. We here provide some descriptive details of the dataset.

| Stats Name | Val |
|---|---|
| Num of Trajs | 200 |
| Total Nav Steps | 10617 |
| Avg Nav Steps | 55.87 |
| Max Traj Len | 130 |
| Min Traj Len | 9 |
| Avg $D_0$ to Goal | 5800 |
| Max $D_0$ to Goal | 13386 |
| Min $D_0$ to Goal | 809 |

### B.2 EVALUATION DATASET

Evaluation are carried out in 3 difficulties: easy, medium, and hard distinguished by the initial Euclidean distance to goal $D_0$. The specific ranges are: $1500 < D_0^{easy} < 4000$, $4000 < D_0^{medium} < 8000$, and $8000 < D_0^{hard}$. Notice success is marked by the agent's Euclidean distance to the goal is within 800. We here provide some descriptive details of the dataset

| Stats Name | Easy | Medium | Hard |
|---|---|---|---|
| Avg $D_0$ | 2673 | 5888 | 9404 |

