# OpenReview forum: "IG-Net: Image-Goal Network for Offline Visual Navigation on A Large-Scale Game Map"
_ICLR.cc/2024/Conference — Submitted to ICLR 2024_

### Official Review · Reviewer_UHPS · 2023-10-30

**Soundness:** 2 fair
**Presentation:** 3 good
**Contribution:** 1 poor
**Rating:** 3
**Confidence:** 5

**Summary:**

This papers presents an agent design named IG-NET for image-goal navigation on large-scale environments.

This agent is trained on an offline learning fashion and relies only on images to navigate.

The model architecture proposed is an image to goal model, trained end-to-end with offline data from experts, with a representation enhancing mechanisms that incorporates position and navigation information prediction.

Experiments show IG-Net robust navigation, being the model that achieves the best performance in all the different difficulty settings tested in the ShooterGame environment.

**Strengths:**

- The paper presents a somehow novel problem: offline image-goal navigation in large-scale game maps.

- Section 5.4 (Case Study) is broad and it illustrates and provides with further evidence to support why IG-Net performs better than the other algorithms evaluated.

**Weaknesses:**

* Section 2 does not provide a clear discussion of why the proposed model (IG-Net) is different from previously published models. It simply mentions that 'IG-Net is meticulously designed to navigate significantly larger maps,' but that is not sufficient.

* Habitat Matterport 3D Semantics Dataset (see missing reference [MR1]) has not been included in the comparison of table 1. This is probably the largest dataset to date, at it should be listed.

* Novelty problems:
  * This is not the first model to propose to solve the navigation problem with offline data. First and foremost, the manuscript overlooks conducting a review of the literature related to imitation learning (behavior cloning) and even recent offline RL, where comparisons with many previous models that have addressed the problem of navigation in virtual environments under the same constraint—using only offline data—could be made.
  * In Section 4.1, no significant differences with respect to previous works are detailed. Previous models used only images as inputs as well [MR2]. There are also works that use positional information to enhance model learning (PIRLNav model does it when using the compass and GPS information as iputs), or even auxiliary tasks (see [MR3]).
  * More works should be included in a detailed discussion: e.g. (Shah et al., 2021) (Kahn et al., 2021) (Kim et al., 2022).


* Clarity needs to be improved:
  * Sections 4.2 - 4.4 need improvement in terms of clarity. Details are missing on how the training of the entire model is structured, and especially, details of the proposed architecture.
  * One cannot use the same symbol for both the model weights and the angles.
  * If the algorithm is trained in an offline manner, which was the offline algorithm used? I could not find any reference about any algorithm.

* Weaknesses of the experimental setup:
  * It is an offline model that just need a small dataset of 200 trajectories. That is not too much. I wonder how this impacts the performance of the model.
  * The way the model is tested does not follow the standard in embodied AI literature. SR is measured when the agent arrives to the target AND samples an stop action, which means that (somehow) it knows that it has arrived. How does the proposed model know that it has arrived to the target? This is not considered in this paper, and the evaluation metrics used do not help to evaluate this fact.
  * More baselines are needed. The selection of VGM is adequate but other approaches have to be considered. Offline-RL models should be considered as well (see [MR4]).
  * How does the proposed approach perform in standard embodied AI navigation benchmarks? Habitat datasets, for example. There is offline data that can be used (see PIRLNav paper dataset of 77k navigation trajectories) or the authors can maybe generate new data (it seems that with just 200 trajectories the IG-Net model has enough information to be trained).




Missing references:
===================
[MR1]   Habitat-Matterport 3D Semantics Dataset.Yadav, Karmesh and Ramrakhya, Ram and Ramakrishnan, Santhosh Kumar and Gervet, Theo and Turner, John and Gokaslan, Aaron and Maestre, Noah and Chang, Angel Xuan and Batra, Dhruv and Savva, Manolis, arXiv preprint , https://arxiv.org/abs/2210.05633, 2022. https://aihabitat.org/datasets/hm3d-semantics/

[MR2] Yuke Zhu, Roozbeh Mottaghi, Eric Kolve, Joseph J. Lim, Abhinav Gupta, Li Fei-Fei, and Ali Farhadi. Target-driven visual navigation in indoor scenes using deep reinforcement learning. In ICRA, 2017.

[MR3] Joel Ye and Dhruv Batra and Abhishek Das and Erik Wijmans. Auxiliary Tasks and Exploration Enable ObjectNav, ICCV, 2022.

[MR4] Offline Reinforcement Learning for Visual Navigation, Dhruv Shah and Arjun Bhorkar and Hrishit Leen and Ilya Kostrikov and Nicholas Rhinehart and Sergey Levine,6th Annual Conference on Robot Learning, 2022.

**Questions:**

I've tried to detail most of the limitations and weaknesses of the proposed model in previous section, with some points that would need to be addressed in a rebuttal.

Overall, I see here a manuscript with incremental contributions. From a theoretical perspective, the proposed model follows a short of imitation learning paradigm with some interesting loss functions that have been specifically designed for the problem of interest. Technically, the proposed architecture is a Masked Auto-Encoder for encoding the images, followed with some embedding concatenations plus positional and action decoding. I can hardly see here a considerable technical contribution either. Overall, the paper describes some ideas that are not adequate, in my humble opinion, for an ICLR conference.

**Details Of Ethics Concerns:**

This work describes a model for creating autonomous agents capable of various tasks: guiding costumers,  building autonomous robots, etc.

The introduced approach  allow to improve the navigation capabilities of autonomous agents. Of course, like most scientific work in the STEM sector, this might lead to some negative societal impacts, which our
work shares with most robotics applications: military robots (used in wrong hands), surveillance etc. The work described in the manuscript was carried out in simulation and as such is unlikely to have produced unethical results, except the impact of large-scale training on CO2 output.

---

> ### Author Response · Authors · 2023-11-23
> **Response to Reviewer UHPS**
>
> To Reviewer UHPS: Thanks for your valuable feedback! We provide the response to each of your questions as follows.
>
> ***Q1: Related work section does not provide a clear discussion of why the proposed model (IG-Net) is different from previously published models***
>
> A1: IG-Net stands out from previous models because it builds an implicit map comprehension through training the model with the several proposed auxiliary tasks: relative and absolute position prediction, distance prediction and navigation path prediction. Specifically, we show the implicit map understanding is especially helpful in large game environments like ShooterGame as compared to VGM, ViNT, and other baselines.
>
> ---
>
> ***Q2: This is not the first model to propose to solve the navigation problem with offline data; Missing related work for offline RL***
>
> A2: We thank your comment for mentioning those related works which we don’t fully cover, and we will add these works in the discussions of our related work part. In the meantime, our method mainly focuses on designing auxiliary tasks for implicit map understanding that benefits the performance for behavior cloning methods, which is different from the main contributions of offline RL algorithms. Also, our method does not leverage reward information, which is in a different setting compared with offline RL methods.
>
> ---
>
> ***Q3: Missing entire model architecture in Section 4.2 - 4.4***
>
> A3: We add the network architecture in Section 4.3 in our revision. Briefly, IG-Net uses MAE-pretrained ViT to separately encode the observation image and goal image, and feed the embedding into four additional transformer layers, and concat the final joint embedding to feed into four different encoders for corresponding tasks.
>
> ---
>
> ***Q4: How the number of trajectories impacts the performance of the model***
>
> A4: We agree that it is important to test the scaling-up ability of our dataset and algorithm, which will be an important part of study in our future work. We also believe that 200 trajectories are sufficient to validate the effectiveness of our proposed approach as our model can successfully reduce all losses on both training and validation sets.
>
> ---
>
> ***Q5: More baselines are needed***
>
> A5: We have added 4 baselines to the experiments including ViNT, GNM, NoMaD, and NoMaD-EMA, shown in Table 2. These baselines all use RGB observations and can be trained without RL. Results show that IG-Net outperforms all the baselines in ShooterGame. We believe the added baselines suffer significant performance loss due to lack of a z-axis, lack of long-horizon training, and lack of a low-level navigator.
>
> ---
>
> ***Q6: How does the proposed approach perform in standard embodied AI navigation benchmarks?***
>
> A6: We have added Gibson experiments where IG-Net is evaluated on different seen and unseen floorplans. Since Gibson environments are much smaller (Table 1), IG-Net doesn’t have an advantage compared to previous approaches. This experiment validates IG-Net has generalization ability to novel environments and the implicit training is especially important in larger game experiments.
>
> ---
>
> ***Q7: No stop action, and performance is not aligned with embodied AI literatures***
>
> A7: In the above experiments on Gibson, we have added the Stop action for both IG-Net and baseline method, and measure the success of navigation only when the agent calls the stop action within the distance threshold of the goal position. We didn’t add Stop action for ShooterGame because this may add an additional factor to the evaluation (the strategy for calling stop action, etc.). However, we will add the Stop action in the future experiments to align with existing literature.
>
> ---
>
> | Setting               | Setting      | Total Trials | SR    | SPL   | DDR   |
> |-----------------------|--------------|--------------|-------|-------|-------|
> | IG-Net                | Gibson-Train | 7200         | **0.539** | **0.471** | **0.447** |
> | IG-Net                | Gibson-Eval  | 1400         | **0.584** | **0.505** | **0.417** |
> | IG-Net (no auxiliary) | Gibson-Train | 7200         | 0     | 0     | 0.323 |
> | IG-Net (no auxiliary) | Gibson-Eval  | 1400         | 0     | 0     | 0.326 |
>
> **Table 1: Generalization capability of IG-Net on Gibson environment on 14 unseen environments.**

---

> ### Author Response · Authors · 2023-11-23
> **Response to Reviewer UHPS (Part 2)**
>
> | Difficulty       |          | Easy     |          |          | Medium   |          |          | Hard     |          |
> |------------------|----------|----------|----------|----------|----------|----------|----------|----------|----------|
> | Metric           | SR       | SPL      | DDR      | SR       | SPL      | DDR      | SR       | SPL      | DDR      |
> | VGM              | 0.24     | 0.09     | 0        | 0        | 0        | 0.3      | 0        | 0        | 0.27     |
> | VGM-Tuned        | 0.2      | 0.09     | 0.39     | 0.08     | 0.05     | 0.39     | 0        | 0        | 0.32     |
> | VGM-Loaded       | 0.32     | 0.14     | 0.49     | 0.04     | 0.02     | 0.26     | 0        | 0        | 0.27     |
> | VGM-Tuned-Loaded | 0.2      | 0.08     | 0.32     | 0.04     | 0.02     | 0.36     | 0.08     | 0.05     | 0.42     |
> | NoMaD            | 0.1      | 0.06     | 0.31     | 0.02     | 0.01     | 0.2      | 0        | 0        | 0.16     |
> | NoMaD-EMA        | 0.08     | 0.05     | 0.27     | 0.06     | 0.04     | 0.26     | 0        | 0        | 0.23     |
> | ViNT             | 0.08     | 0.07     | 0.17     | 0.02     | 0.02     | 0.16     | 0        | 0        | 0.15     |
> | GNM              | 0.04     | 0.04     | 0.13     | 0.04     | 0.03     | 0.15     | 0.02     | 0.02     | 0.14     |
> | IG-Net           | **0.54** | **0.24** | **0.75** | **0.26** | **0.17** | **0.65** | **0.24** | **0.15** | **0.75** |
> | IG-Net noaux     | 0.18     | 0.09     | 0.36     | 0.14     | 0.08     | 0.42     | 0        | 0        | 0.44     |
>
> **Table 2: Navigation Performance of IG-Net compared with NoMaD, ViNT, and GNM.**

---

### Official Review · Reviewer_rCRb · 2023-10-31

**Soundness:** 2 fair
**Presentation:** 3 good
**Contribution:** 3 good
**Rating:** 6
**Confidence:** 4

**Summary:**

The paper presents the Image-Goal Network (IG-Net), a novel approach to address the challenges of navigating large-scale gaming environments with limited offline data. The model is trained end-to-end, incorporating position, path, and distance prediction for implicit spatial mapping. Experiments in the ShooterGame environment validate IG-Net's efficacy, and the paper also contrasts it with existing methods in visual navigation.

**Strengths:**

1. The authors present an interesting and amazing work in the paper and raise the meaningful challenge of navigation in large-scale environments to the community.
3. The authors introduce an expansive environment ShooterGame environment. The ShooterGame environment is nearly 20 times larger than the previous environment on one map.
3. IG-Net utilizes a variety of auxiliary tasks, including local and global path planning and navigation distance prediction, to strengthen its visual navigation capabilities. These auxiliary tasks enrich the agent's representation and improve its navigation performance.
2. In comparison to other methodologies like VGM, IG-Net outperforms in terms of success rate, success weighted by path length, and navigation efficiency. It achieves higher success rates and navigation efficiency even under more challenging settings.

**Weaknesses:**

1. The authors perform the testing on the seen environment during training. I believe that demonstrating the performance in unseen environments will consolidate the work.
2. The authors do not provide the details for the architecture of the IG-Net. I am a little bit confused about the input/output of each network component.
3. The ablation study in Table 4 shows an experiment "IG-Net (no position)." I am curious that the experiment demonstrates the performance of IG-Net without both absolute/relative position prediction or either of them. A similar question about the ablation experiment "IG-Net (no path and dist)", why did the authors remove two auxiliary tasks at the same time in the ablation study?

**Questions:**

1. Although the coverage of the ShooterGame environment is much larger than other environments, it also seems to be more empty than other environments. Could the authors discuss the difficulty differences in navigation among these environments except the coverage area? Meanwhile, I am curious about the number of different states in the ShooterGame environment.
1. How do the authors concatenate the encoding embeddings from the first step into a unified representation? Is that a FIFO stack or an MLP/pooling layer serving as a projector to ensure the same size of concatenated features during different lengths of episodes?
2. In the third part of the architecture design, the authors mentioned "Utilizing position, path, distance, and action decoders. " I am curious whether the historical information, apart from visual embedding, is included in the action prediction. If the authors do not use historical action/state representations, how does the agent maintain the search efficiency instead of being stuck in a small area?
4. Suppose we put the trained agent in an unseen environment, how will the agent perform? In the unseen environment, will the agent still be able to perform these auxiliary tasks? How does the agent perform on these auxiliary tasks, e.g., absolute/relative position prediction?

---

> ### Author Response · Authors · 2023-11-23
> **Response to Reviewer rCRb**
>
> To Reviewer rCRb: Thanks for your valuable feedback! We provide the response to each of your questions as follows.
>
> ***Q1: Although being larger, ShooterGame is more empty. Discuss other metrics for navigation difficulty.***
>
> A1: Other works use metrics including specific surface area and navigation complexity to show the complexity of the environment (see Table 1 in the paper of Gibson dataset https://arxiv.org/pdf/1808.10654.pdf). We believe that these two metrics overlook the scale of the environment which greatly affects the navigation difficulty in the environment. We think that the minimum angle of turns alongside the navigation episode is a great metric. We will add more metrics for comparing navigation complexity in our future work.
>
> ---
>
> ***Q2: The paper doesn’t provide network architecture for IG-Net, and input/output for the network***
>
> A2: We add the network architecture in Section 4.3 in our revision. Briefly, IG-Net uses MAE-pretrained ViT to separately encode the observation image and goal image, and feed the embedding into four additional transformer layers, and concat the final joint embedding to feed into four different encoders for corresponding tasks.
>
> ---
>
> ***Q3: IG-Net trains and tests in the same environment. Should demonstrate results in novel environments***
>
> A3: Please refer to our new experiments on Gibson, where the agent is evaluated on 14 unseen environments for short horizon navigation.  Results show that IG-Net can navigate well in unseen environments and outperforms no auxiliary task baselines, showing the generalizability of IG-Net.
>
> ---
>
> ***Q4: The definition of “No position”, and why removing two auxiliary tasks at the same time***
>
> A4: “No position” ablation study simultaneously removes both relative position prediction and absolute position prediction. We remove two auxiliary tasks at the same time to verify the joint effect of position loss as a whole. We have realized that removing each single auxiliary task for ablation study is better and will perform the experiment in our future work.
>
> ---
>
> ***Q5: Whether historical states serve as inputs for IG-Net. Historical states are crucial to searching behaviors***
>
> A5: We really appreciate the comment. Currently, IG-Net only uses one current image and one goal image as inputs, which we also observe failure searching behavior around the corner of the map and hope to try adding historical states in the inputs. In the future work, we will carefully design experiments to verify the design for historical states.

---

### Official Review · Reviewer_Sxb4 · 2023-11-06

**Soundness:** 2 fair
**Presentation:** 2 fair
**Contribution:** 2 fair
**Rating:** 5
**Confidence:** 4

**Summary:**

The paper proposes IG-Net, a end-to-end method designed for visual navigation in large scale environments like the shooter game environment from unreal engine. Instead of building an explicit map IG-Net leverages visual and implicit pose information to learn better representations for navigation. To do this IG-Net uses a combination of relative pose prediction, absolute pose prediction, navigation distance and navigation path prediction as auxiliary tasks on offline trajectories and randomly sampled pose and visual data. The paper also shows IG-Net outperforms a Visual-Graph-Memory baseline which uses topological graph representation in the ShooterGame environment.

**Strengths:**

1. The paper proposes a method to test Image-Goal Navigation in large-scale game environments which require long-horizon navigation skills
2. IG-Net outperforms VGM baseline by a significant margin on all 3 evaluation splits (easy, medium, hard)
3. Ablation in table 2 nicely show the importance of auxiliary objectives in boosting evaluation performance for IG-Net
4. Experiments in section 5.4.2. show that IG-Net is also robust to noise to some extent and doesn’t lead to significant drop in performance on the hard split. But for some other splits we do see a large drop in performance.
5. Paper is easy to follow

**Weaknesses:**

1. The experiments section doesn’t show comparison with state-of-the-art methods on ImageNav. There has been some recent works in ImageNav which use end-to-end RL training using pretrained representations (visual and pose) [1, 2, 3, 4] which achieve significant performance on the ImageNav benchmark. The current state-of-the-art method [1] acheives ~92% SR using simple concatenation of goal and observation image being fed to a single ResNet model to generate a observation embedding. OVRL [3] and OVRL-v2 [2] have been using pretrained visual representations using data from omnidata which leads to improved perfomance. It is essential that the authors compare IG-Net to the state-of-the-art baselines. I’d suggest adding comparison to FGPrompt-EF baseline from [1] and OVRL and OVRL-v2 pretrained baselines from [2, 3]. [4] seems to be a concurrent work with no code released so no need to add comparison with it.
2. To the best of my understanding, the experimental setup is not testing generalization to new maps in the ShooterGame environment. Can authors please show results of generalization to different maps in the environment? If the performance is being tested in the same scene then both methods should be able to achieve significantly higher success. Can authors please share more details about the VGM baseline’s training and evaluation setup? In addition, it would be good if the authors can add more details about the ShooterGame dataset they used for training and testing.
3. Based on description in section 4.3 it is unclear what the policy architecture of IG-Net looks like. Can authors please share a policy architecture figure?
4. Ablations in section 5.5 need to ablate each component of auxiliary task separately. In the current results it looks like row 2 only uses relative and absolute pose prediction and row 3 doesn’t use navigation path prediction and distance aux tasks. In each ablation authors should sequentially remove every single loss from list of auxiliary losses to clearly demonstrate effectiveness of each components.

[1] X. Sun, P. Chen, J. Fan, T. Li,… FGPrompt: Fine-grained Goal Prompting for Image-goal Navigation. In NeurIPS, 2023
[2] K. Yadav, A. Majumdar, R. Ramrakhya, N. Yokoyama, A. Baevski, Z. Kira, O. Maksymets, and D. Batra. Ovrl-v2: A simple state-of-art baseline for imagenav and objectnav. arXiv preprint arXiv:2303.07798, 2023.
[3]  K. Yadav, R. Ramrakhya, A. Majumdar, V.-P. Berges, S. Kuhar, D. Batra, A. Baevski, and O. Maksymets. Offline visual representation learning for embodied navigation. In International Conference on Learning Representations (ICLR), 2022.
[4] G. Bono, L, Antsfeld, B.  Chidlovskii, P. Weinzaepfel, C. Wolf End-to-End, (Instance)-Image Goal Navigation Through Correspondence as an Emergent Phenomenon, arxiv preprint arxiv:2309.16634

**Questions:**

1. The best methods [1,2,3,4] on ImageNav task use RL for training. It would be good if authors can add comparison with a RL trained baseline using FGPrompt-EF. Can authors provide more details into why it is not possible to get online data? Is it due to slow simulator speed?

---

> ### Author Response · Authors · 2023-11-23
> **Response to Reviewer Sxb4**
>
> To Reviewer Sxb4: Thanks for your valuable feedback! We provide the response to each of your questions as follows.
>
> ***Q1: Compare with state-of-the-art methods on ImageNav***
>
> A1: We have included comparison between ViNT, GNM, and NoMaD baselines, shown in Table 2. We thank the reviewers for suggesting SOTA methods such as FGPrompt and OVRL. However, to keep the comparison fair and due to the restrictiveness of our environment in simulation speed and perception domains, we have to focus on non-RL algorithms that use RGB observations only.
>
> ---
>
> ***Q2: The experimental setup is not testing generalization to new maps in the ShooterGame environment***
>
> A2: We perform additional experiments of IG-Net on new maps in Gibson experiments, where the agent is trained on 396 environments and evaluated on 14 novel ones, shown in Table 1. Results show that IG-Net can navigate well in the easy setting and outperforms no auxiliary task baselines, showing the generalizability of IG-Net.
>
> ---
>
> ***Q3: Unclear architecture for policy network***
>
> A3: We add the network architecture in Section 4.3 in our revision. Briefly, IG-Net uses MAE-pretrained ViT to separately encode the observation image and goal image, and feed the embedding into four additional transformer layers, and concat the final joint embedding to feed into four different encoders for corresponding tasks.
>
> ---
>
> ***Q4: Ablate each auxiliary task separately***
>
> A4: We really appreciate the comment, and will perform the experiment for ablating each auxiliary task in our future revision of this work.
>
> ---
>
> ***Q5: Why it is not possible to get online data for RL approach***
>
> A5: We thank the reviews for suggesting the RL approach. Currently, our simulator runs at a maximum of 5 FPS to wait until an action is executed through keyboard simulation. This is too slow for RL training compared to 62.5 FPS in MuJoCo [1] and 8000-10000 FPS in Habitat-Sim [3][4].
>
> ---
>
> [1] MuJoCo Official Documentation: https://mujoco.readthedocs.io/en/stable/programming/samples.html?highlight=fps#sabasic
>
> [2] Emanuel Todorov, Tom Erez, and Yuval Tassa. Mujoco: A physics engine for model-based control. In 2012 IEEE/RSJ International Conference on Intelligent Robots and Systems, pp. 5026–5033, 2012. doi: 10.1109/IROS.2012.6386109.
>
> [3] Habitat-Sim Official Documentation: https://aihabitat.org/#:~:text=Habitat%2DSim%20simulates%20a%20Fetch,body%20dynamics%20for%201%2F30sec.
>
> [4] Manolis Savva*, Abhishek Kadian*, Oleksandr Maksymets*, Yili Zhao, Erik Wijmans, Bhavana Jain, Julian Straub, Jia Liu, Vladlen Koltun, Jitendra Malik, Devi Parikh, and Dhruv Batra. Habi-tat: A Platform for Embodied AI Research. In Proceedings of the IEEE/CVF International Conference on Computer Vision (ICCV), 2019.
>
> | Setting               | Setting      | Total Trials | SR    | SPL   | DDR   |
> |-----------------------|--------------|--------------|-------|-------|-------|
> | IG-Net                | Gibson-Train | 7200         | **0.539** | **0.471** | **0.447** |
> | IG-Net                | Gibson-Eval  | 1400         | **0.584** | **0.505** | **0.417** |
> | IG-Net (no auxiliary) | Gibson-Train | 7200         | 0     | 0     | 0.323 |
> | IG-Net (no auxiliary) | Gibson-Eval  | 1400         | 0     | 0     | 0.326 |
>
> Table 1: Generalization capability of IG-Net on Gibson environment on 14 unseen environments.
>
> | Difficulty       |          | Easy     |          |          | Medium   |          |          | Hard     |          |
> |------------------|----------|----------|----------|----------|----------|----------|----------|----------|----------|
> | Metric           | SR       | SPL      | DDR      | SR       | SPL      | DDR      | SR       | SPL      | DDR      |
> | VGM              | 0.24     | 0.09     | 0        | 0        | 0        | 0.3      | 0        | 0        | 0.27     |
> | VGM-Tuned        | 0.2      | 0.09     | 0.39     | 0.08     | 0.05     | 0.39     | 0        | 0        | 0.32     |
> | VGM-Loaded       | 0.32     | 0.14     | 0.49     | 0.04     | 0.02     | 0.26     | 0        | 0        | 0.27     |
> | VGM-Tuned-Loaded | 0.2      | 0.08     | 0.32     | 0.04     | 0.02     | 0.36     | 0.08     | 0.05     | 0.42     |
> | NoMaD            | 0.1      | 0.06     | 0.31     | 0.02     | 0.01     | 0.2      | 0        | 0        | 0.16     |
> | NoMaD-EMA        | 0.08     | 0.05     | 0.27     | 0.06     | 0.04     | 0.26     | 0        | 0        | 0.23     |
> | ViNT             | 0.08     | 0.07     | 0.17     | 0.02     | 0.02     | 0.16     | 0        | 0        | 0.15     |
> | GNM              | 0.04     | 0.04     | 0.13     | 0.04     | 0.03     | 0.15     | 0.02     | 0.02     | 0.14     |
> | IG-Net           | **0.54** | **0.24** | **0.75** | **0.26** | **0.17** | **0.65** | **0.24** | **0.15** | **0.75** |
> | IG-Net noaux     | 0.18     | 0.09     | 0.36     | 0.14     | 0.08     | 0.42     | 0        | 0        | 0.44     |
>
> Table 2: Navigation Performance of IG-Net compared with NoMaD, ViNT, and GNM.

---

### Official Review · Reviewer_JFQf · 2023-11-07

**Soundness:** 2 fair
**Presentation:** 3 good
**Contribution:** 2 fair
**Rating:** 3
**Confidence:** 4

**Summary:**

The paper proposes a method of training navigation agents which can navigate large spaces using ego-centric RGB observations without assuming access to accurate position information. The task is to navigate to an image-goal in a large simulated environment. The method utilizes several auxiliary losses to learn the spatial map and how to navigate it implicitly. The authors demonstrate their method on several trajectories of varying difficulty in a single environment.

**Strengths:**

- The paper studies navigating large-spaces using RGB egocentric observations without accurate localisation which is an interesting problem that hasn’t been extensively studied before.
- The environment studied in the setup is dynamic which makes the task even more challenging (with varying lighting conditions, moving objects and other distractors)
- The authors train an end-to-end approach for navigating in large spaces which hasn’t been shown to work before.  They compare their method against meaningful baselines (baselines that build an explicit representation of the environment as graphs) and show superior results

**Weaknesses:**

- The authors test their method on a single environment which is a fairly restrictive experimental setup. We don’t know if the model is capable of handling / memorising multiple environments. I would have liked to see experiments in which a single model can be learned for multiple environments.
- The experiments also assume perfect actuation — left/right and forward will always lead to the same amount of actuation. I wonder how does the agent behave when the actuation during test-time is noisy / different from those used during training. Such a situation is fairly common in the real world and it would be nice to show experiments for the more general setting.
- More importantly, the paper doesn’t have any analysis and discussion about the train-time vs test-time distribution. How different are the trajectories used at train-time compared to test-time trajectories. This analysis is important to understand the model’s generalisation capability and to actually test if it has build an efficient implicit representation of the map or not.
- I would have liked to see some more experiments to probe this implicit understanding of the map. For instance, if the agent has only been trained on a complex trajectory, can it learn to output the shortest path between two pair of observations?
- Similar to the last point, I would have also liked to see the effect of distractors in the observation on task performance. For instance, does the performance decrease when we add more distractors in the observation at test time? Since one of the novelties of this task setup is dynamic observations, it feels that this kind of analysis is important for a more thorough evaluation.

**Questions:**

- Not a question, but a comment. The paper overloads the meaning of \theta in equation (2) and (3) which made parsing those equations slightly confusing. I would also encourage the authors to run a more extensive grammar check to improve the writing of the paper.

- According to my understanding, none of the losses in Section 4.2 actually involves predicting actions. How does the method learn to predict actions?

- Related to weaknesses:
    - it’d be great to study the effect of agent’s observation field of view and  different action spaces on task performance.
    - it’d be interesting to study the robustness of the method when action spaces change during test time
    - can the authors comment on how different test time trajectories are to the ones seen during training? For example, for each location of the agent for a trajectory at test time, what is the nearest location observed during training?

---

> ### Author Response · Authors · 2023-11-23
> **Response to Reviewer JFQf**
>
> To Reviewer JFQf: Thanks for your valuable feedback! We provide the response to each of your questions as follows.
>
> ***Q1: Experiments with a single model learned for multiple environments***
>
> A1: We perform additional experiments of IG-Net on new maps in Gibson experiments, where the agent is trained on 396 environments and evaluated on 14 novel ones, shown in Table 1. Results show that IG-Net can navigate well in the easy setting and outperforms no auxiliary task baselines, showing the generalizability of IG-Net.
>
> ---
>
> ***Q2: Performance when observation and actuation noise exists, which is common in the real world***
>
> A2: Actuation noise is a very important problem for real world applications, and it can be simulated in game engines by adding action noise. Exploring how IG-Net can handle actuation noise is a good research direction and we will explore it in future work. Observation noise requires specific design in the camera, and we will test the performance in the future work.
>
> ---
>
> ***Q3: Show train-time and test-time state-goal distribution***
>
> A3: We verify the difference of test-time state-goal pair and training state-goal pair by defining a metric of minimum state-goal distance. For each evaluation state-goal pair $(s,g)$, the minimum state-goal distance is defined to be $\min_{i} \|s_i - s\|_2 + \|g_i - g\|_2$, where $(s_1, g_1), …, (s_N, g_N)$ are $N=200$ state-goal pairs for navigation training data. Minimum state-goal distance describes the lowest difference in the training dataset of an evaluation state-goal pair. The mean value of minimum state-goal distance on the whole evaluation data is $2130$ on $D<4000$ difficulty, $2260$ on the $4000<D<8000$ difficulty, and $2375$ on the $D>8000$ difficulty, which is much larger than the distance threshold for success judgment ($800$). This demonstrates that the evaluation state-goal pairs have a small overlap with the training dataset.
>
> ---
>
> ***Q4: Experiment for probing the implicit understanding of the map, e.g. predicting shortest path***
>
> A4: This is a great direction for testing the internal scene understanding in the IG-Net. We will perform detailed analysis on designing probing tasks for pretrained IG-Net to verify this in the future work.
>
> ---
>
> ***Q5: How does IG-Net perform action predictions***
>
> A5: Actions are categorical {Forward, Left, Right} and are learned through Cross-Entropy loss between expert training data. We have added the description for action prediction in Section 4.2 in our revisions.
>
> ---
>
> | Setting               | Setting      | Total Trials | SR    | SPL   | DDR   |
> |-----------------------|--------------|--------------|-------|-------|-------|
> | IG-Net                | Gibson-Train | 7200         | **0.539** | **0.471** | **0.447** |
> | IG-Net                | Gibson-Eval  | 1400         | **0.584** | **0.505** | **0.417** |
> | IG-Net (no auxiliary) | Gibson-Train | 7200         | 0     | 0     | 0.323 |
> | IG-Net (no auxiliary) | Gibson-Eval  | 1400         | 0     | 0     | 0.326 |
>
> **Table 1: Generalization capability of IG-Net on Gibson environment on 14 unseen environments.**

---

### Official Review · Reviewer_m9pY · 2023-11-07

**Soundness:** 2 fair
**Presentation:** 2 fair
**Contribution:** 2 fair
**Rating:** 3
**Confidence:** 4

**Summary:**

This paper deals with the problem of Image Goal tasks when agent pose in environment is unknown, and shows how training with limited navigation data (but good amount of positional data) in large scale game space like areas -  is still possible to yield good results. The authors have introduced a Network archiecture to represent spatial map information implicitly based on prediction of position, distance and path. Although the application setup is novel, but the paper needs an overhaul to bring in the novel aspects and add technical challenges that were overcome. The ablation studies and experiments do support the writings.

**Strengths:**

The usage of distance decrement rate (DDR) is logical in this paper wrt task, however a scaling may help in evaluations.
The code is available and executed as mentioned.
The scope of application is good and will open up research communities to tackle the problem with game to real transfer later.

**Weaknesses:**

In Fig. 1, the separating boundary walls are quite distant due to the open space, making exploration easy compared to close-looped indoor environment with full or partial occlusions.
The SPL compared to success rate in Table 2 section 1, is not that great - any logic for that? Although a gap in the metrics will help the future researchers to improve the SOTA.
The architecture of the work should be clearly highlighted and mentioned.
The video is not clear in terms of what is being planned to be achieved - this needs serious improvement - like how the scene is processed and the action outputs wrt image goal is coming.
More SOTA study and real world transfer is expected.

**Questions:**

Not sure how much the trained policy is suitable for FPS games compared to real world robotic deployments in the wild in human co-occupied spaces.
The action space should be clearly mentioned for the testbed. Also how dependent on camera intrinsics.
Incorporation of Auxiliary Tasks - how is it done - need finer details.

---

> ### Author Response · Authors · 2023-11-23
> **Response to Reviewer m9pY**
>
> To Reviewer m9pY: Thanks for your valuable feedback! We provide the response to each of your questions as follows.
>
> ***Q1: Boundary walls are distant so exploration is easy compared to indoor environments***
>
> A1: Although the boundary walls are relatively distant, the small width of the corridors compared to the large game map still poses great challenges for exploration.
>
> ---
>
> ***Q2: SPL is not good compared to SR***
>
> A2: We mention the reason for this in Section 5.2. We use Euclidean distance between state and goal position since geodesic distance is not available in ShooterGame, which makes SPL smaller compared to normal cases. We should more clearly show this fact in the table. We may also use expert path length as comparison for SPL calculation to alleviate the problem.
>
> ---
>
> ***Q3: Network architecture should be more clear***
>
> A3: We add the network architecture in Section 4.3 in our revision. Briefly, IG-Net uses MAE-pretrained ViT to separately encode the observation image and goal image, and feed the embedding into four additional transformer layers, and concat the final joint embedding to feed into four different encoders for corresponding tasks.
>
> ---
>
> ***Q4: How much is the policy suitable for FPS games compared to real world robots***
>
> A4: Our challenge setting mimics the real-world human player perception in games. By not requiring a panoramic camera, depth camera, or global pose, we showcase the robustness of our navigation algorithm. We do agree that there is a domain gap between game simulators and real-world when humans and other agents are involved, which is a different type of setting than this paper.
>
> ---
>
> ***Q5: Clearly mention the action space***
>
> A5: Our action is discrete and similar to how humans play games using a keyboard. Specifically, the actions are {Forward, Turn Left, Turn Right} with key-pressing time being {0.4, 0.2, 0.2} seconds, respectively. We use UnrealCV as the APIs for executing the actions in the unreal engine simulator.

---

### Author Response · Authors · 2023-11-23
**General Response to all Reviewers**

We thank all reviewers for your insights on our paper. We here address the shared concerns of reviewers and summarize the important components added to the paper.

- To answer questions regarding “multiple map generalizability”, we have added experiments of IG-Net on Gibson Easy. Our model is trained on 396 different floor plans, evaluated on 72 from the training set and 14 unseen ones in the testing set. Results below show that IG-Net can generalize across multiple environments and solve unseen maps. We further point out that the key contribution of our work is proposing auxiliary training tasks to build an implicit map comprehension, which is in most necessity for large-scale maps like ShooterGame. We therefore didn’t focus on generalizability in our original submission.

- To address concerns regarding “more baselines are needed”, we have added 4 baselines (ViNT [1], GNM [2], NoMaD [3], NoMaD-EMA). We thank reviews for suggesting other recent related works. However, constrained by the low simulation FPS and limited perception in ShooterGame, we cannot train RL baselines or baselines require more than RGB vision.

- We have added the action prediction section in the updated paper Section 4.2, and a model architecture image in the updated paper Section 4.3.

[1] Dhruv Shah, Ajay Sridhar, Nitish Dashora, Kyle Stachowicz, Kevin Black, Noriaki Hirose, and Sergey Levine. Vint: A foundation model for visual navigation. CoRR, abs/2306.14846, 2023.

[2] Ajay Sridhar, Dhruv Shah, Catherine Glossop, and Sergey Levine. NoMaD: Goal Masked Diffusion Policies for Navigation and Exploration. arXiv pre-print, 2023. URL https://arxiv.org/abs/2310.07896.

[3] Dhruv Shah, Ajay Sridhar, Arjun Bhorkar, Noriaki Hirose, and Sergey Levine. GNM: A General Navigation Model to Drive Any Robot. In International Conference on Robotics and Automation (ICRA), 2023a. URL https://arxiv.org/abs/2210.03370.

| Setting               | Setting      | Total Trials | SR    | SPL   | DDR   |
|-----------------------|--------------|--------------|-------|-------|-------|
| IG-Net                | Gibson-Train | 7200         | **0.539** | **0.471** | **0.447** |
| IG-Net                | Gibson-Eval  | 1400         | **0.584** | **0.505** | **0.417** |
| IG-Net (no auxiliary) | Gibson-Train | 7200         | 0     | 0     | 0.323 |
| IG-Net (no auxiliary) | Gibson-Eval  | 1400         | 0     | 0     | 0.326 |

**Table 1: Generalization capability of IG-Net on Gibson environment on 14 unseen environments.**

| Difficulty       |          | Easy     |          |          | Medium   |          |          | Hard     |          |
|------------------|----------|----------|----------|----------|----------|----------|----------|----------|----------|
| Metric           | SR       | SPL      | DDR      | SR       | SPL      | DDR      | SR       | SPL      | DDR      |
| VGM              | 0.24     | 0.09     | 0        | 0        | 0        | 0.3      | 0        | 0        | 0.27     |
| VGM-Tuned        | 0.2      | 0.09     | 0.39     | 0.08     | 0.05     | 0.39     | 0        | 0        | 0.32     |
| VGM-Loaded       | 0.32     | 0.14     | 0.49     | 0.04     | 0.02     | 0.26     | 0        | 0        | 0.27     |
| VGM-Tuned-Loaded | 0.2      | 0.08     | 0.32     | 0.04     | 0.02     | 0.36     | 0.08     | 0.05     | 0.42     |
| NoMaD            | 0.1      | 0.06     | 0.31     | 0.02     | 0.01     | 0.2      | 0        | 0        | 0.16     |
| NoMaD-EMA        | 0.08     | 0.05     | 0.27     | 0.06     | 0.04     | 0.26     | 0        | 0        | 0.23     |
| ViNT             | 0.08     | 0.07     | 0.17     | 0.02     | 0.02     | 0.16     | 0        | 0        | 0.15     |
| GNM              | 0.04     | 0.04     | 0.13     | 0.04     | 0.03     | 0.15     | 0.02     | 0.02     | 0.14     |
| IG-Net           | **0.54** | **0.24** | **0.75** | **0.26** | **0.17** | **0.65** | **0.24** | **0.15** | **0.75** |
| IG-Net noaux     | 0.18     | 0.09     | 0.36     | 0.14     | 0.08     | 0.42     | 0        | 0        | 0.44     |

**Table 2: Navigation Performance of IG-Net compared with NoMaD, ViNT, and GNM.**

---

### Meta-Review · Area_Chair_8Jkx · 2023-12-09

**Metareview:**

The paper presents an approach for Image Goal Navigation in large scale environments. In particular, it studies this problem from first person observations with limited localization information. It proposes to use auxiliary losses related to predicting position information as well as an implicit map.

The main strengths of the paper are addressing an important problem of large scale navigation. They find the performance on the single eval environment good, and the presented ablation analysis w.r.t some of the algorithmic choices convincing.

The main weaknesses of the originally submitted paper is unconvincing experiments. In particular, the authors present test results on a single environment, the one they have trained on. Further, they do not compare adequately as a result. Upon rebuttal the authors present results on Gibson, that aren't competitive when compared to other approaches. Some reviewers find the need for additional analysis of the approach with respect to more realistic settings, e.g. noise in actuations etc.

**Justification For Why Not Higher Score:**

The paper received 3 x reject, 1 x marginally reject and 1 x marginally accept. Although the reviewers find the problem settings important and the approach promising as evaluated in the paper, upon a more thorough evaluation on established settings they aren't convinced that it is competitive. Hence, the paper is rejected from ICLR 2024.

We recommend to the authors to provide a more convincing experimental analysis on this established problem.

**Justification For Why Not Lower Score:**

N/A

---

### Decision · Program_Chairs · 2024-01-16

Reject